# Clinical Role of Pharmacists in the Care of Incarcerated People at Correctional Facilities: A Scoping Review

**DOI:** 10.3390/pharmacy13050113

**Published:** 2025-08-24

**Authors:** Christian Eduardo Castro Silva, Beatriz Bernava Sarinho, Michelle Bonafé, Tácio de Mendonça Lima, Inajara Rotta, Samara Jamile Mendes, Patricia Melo Aguiar, Marília Berlofa Visacri

**Affiliations:** 1Department of Pharmacy, School of Pharmaceutical Sciences, University of São Paulo, São Paulo 05508-000, SP, Brazil; christian.castro.cs@usp.br (C.E.C.S.); beatriz.sarinho@alumni.usp.br (B.B.S.); michelle.bonafe@usp.br (M.B.); samarajm@usp.br (S.J.M.); aguiar.pm@usp.br (P.M.A.); 2Department of Pharmacy and Pharmaceutical Administration, Faculty of Pharmacy, Federal Fluminense University, Niterói 24241-000, RJ, Brazil; taciolima@id.uff.br; 3Department of Pharmacy, Federal University of Paraná, Curitiba 80210-170, PR, Brazil; inajara.rotta@ufpr.br

**Keywords:** incarcerated population, pharmacist, pharmaceutical services, review

## Abstract

This study aimed to map the literature on the clinical role of pharmacists in the care of incarcerated people at correctional facilities and to identify gaps in this field. A scoping review was conducted on 30 July 2024, using the PubMed, Scopus, and LILACS databases. Gray literature was searched via Google Scholar, and references of included studies were manually reviewed. Primary studies of any design reporting pharmacists’ clinical services and/or activities for incarcerated individuals were eligible. Study selection and data extraction were performed independently by two reviewers, with a third resolving disagreements. The search yielded 894 records, from which 27 studies were included. Most studies were conducted in the United States (*n* = 16; 59%) and France (*n* = 7; 26%). Eleven (41%) focused exclusively on male populations, and one (4%) on female inmates. Most studies addressed pharmacists’ clinical roles in mental health conditions and substance use disorders (*n* = 9; 33%), infectious diseases (*n* = 5; 19%), and diabetes (*n* = 4; 15%). Clinical services and/or activities related to direct patient care were the most frequently reported (*n* = 18; 67%). Process measures were reported in 18 studies (67%), and clinical outcomes were the most common type of outcome (*n* = 13; 48%). This review highlights the pharmacist’s clinical role in treating mental health conditions and substance abuse, infectious diseases, and diabetes in incarcerated care. It underscores the need for further research in low- and middle-income countries, on women’s health, and on other prevalent conditions.

## 1. Introduction

Globally, around 11 million people are incarcerated, with the largest populations found in the United States (1,808,100), China (1,690,000), and Brazil (850,377) [1]. The incarcerated population has significantly increased across almost every continent in the last twenty years [1]. This population is socially vulnerable, as many prisoners face inadequate living conditions in facilities with structural problems, overcrowding, low-quality or insufficient food, and limited access to healthcare [2,3]. These factors increase their susceptibility to developing diseases or exacerbating pre-existing conditions [3]. The right to health must be guaranteed to this population, regardless of the crimes committed, as it is fundamentally a human rights issue [3].

Compared to the general population, the incarcerated population in correctional facilities is more vulnerable to various infectious and communicable diseases, including tuberculosis, viral hepatitis, syphilis, and HIV/AIDS [4,5]. Additionally, many individuals are diagnosed with chronic conditions such as hypertension, diabetes, and mental health conditions [4]. Substance use disorders and withdrawal also frequently affect prisoners [4,5]. In light of this, the American Society of Health-System Pharmacists (ASHP) recommends that correctional facilities provide at least a basic, humane, and adequate level of healthcare services, accessible to inmate-patients 24 h a day [6].

Despite policy efforts, correctional facilities often have underdiagnosed patients, leading to untreated and uncontrolled health conditions [7]. Adding to these challenges, limited access to appropriate treatments is also a reality. Studies reveal an underuse of medications in this population, ranging from 1.9 to 5.5 times less, depending on the condition, compared to the general population [8,9]. Due to this scarcity, some patients tend to seek alternative ways to obtain medications, such as relying on family members, which can potentially lead to a lack of knowledge among the healthcare team regarding their treatments. There are still patients who are unable to access medications through alternative ways, resulting in going without medication, adversely affecting their health outcomes [10].

Pharmacists, as integral members of the healthcare team, can provide pharmacist services in various settings, particularly for vulnerable populations, including those incarcerated. It is recommended that all correctional facilities secure the services of a pharmacist [6]. However, the presence of pharmacists in correctional facilities remains limited in some settings, particularly in underfunded correctional facilities or in low- and middle-income countries [11]. When pharmacists are present, they encounter numerous challenges. Pharmacists often have limited contact with incarcerated individuals and must work in pharmacy facilities with poor infrastructure and scarce resources, including insufficient access to necessary medications. These limitations further complicate their ability to perform daily duties effectively [11].

Thus, pharmacists are frequently restricted to technical and logistical responsibilities [12]. Despite these limitations, pharmacists remain essential contributors to the patient-centered care of the incarcerated population. Correctional pharmacists could be dedicated to helping inmate-patients achieve optimal health outcomes while minimizing the risk of harm [6]. To the best of our knowledge, only one narrative literature review has explored the roles of pharmacists in correctional facilities. This review indicated that pharmacists acting as primary care providers in this setting may engage in activities such as direct patient care, participation in healthcare clinics, and medication management, based on data published up to 2017 [12]. Since then, a substantial body of literature has emerged, and it is important to map these new findings to assess whether the clinical role of pharmacists in this context has evolved. In addition, this scoping review aims to identify, where available, outcomes that demonstrate the potential benefits of integrating pharmacists into healthcare teams caring for incarcerated populations, as well as best practices that could be replicated to optimize their clinical contributions. The chosen study design supports not only the synthesis of existing evidence but also the identification of knowledge gaps that may guide future research.

## 2. Materials and Methods

### 2.1. Study Design

A scoping review was carried out following the recommendations of the Joanna Briggs Institute (JBI) Reviewer’s Manual [13] and reported according to the criteria of the Preferred Reporting Items for Systematic reviews and Meta-Analyses statement for Scoping Reviews (PRISMA-ScR) [14]. The protocol for this review is available upon request from the corresponding author.

### 2.2. Review Question

The following research question was formulated to guide this scoping review: What research has been published on the clinical services and/or activities performed by pharmacists in correctional facilities, and what are the main outcomes and process measures reported among incarcerated individuals?

### 2.3. Search Strategies

A comprehensive literature search was conducted in the PubMed, Scopus, and Latin American and Caribbean Health Sciences Literature (LILACS) databases for studies published from the inception of the database until 30 July 2024. Additionally, the gray literature was explored through Google Scholar (limited to 100 entries, excluding patents and citations). The complete search strategies for all databases are detailed in Appendix A. Furthermore, references from all included articles were reviewed to identify any studies that may have been missed.

### 2.4. Eligibility Criteria

Eligibility criteria were established based on the Population, Concept, and Context framework: (a) Population: incarcerated people; (b) Concept: clinical services and/or activities provided by pharmacists; (c) Context: correctional facilities.

Articles were included if they addressed clinical services and/or activities provided by pharmacists to incarcerated individuals in any type of custodial correctional facility, regardless of whether they reported process measures or outcomes data. Survey studies, books/book chapters, dissertations and theses, editorials, conference proceedings or abstracts, the literature reviews, and guidelines were excluded. Articles unavailable or published in non-Roman characters were also excluded.

### 2.5. Study Selection

The manuscripts retrieved from the databases were allocated to the Rayyan QCRI web program [15] to exclude duplicate files, analyze the titles and abstracts of the articles, and analyze complete articles whose abstracts were previously selected. The initial screening of titles and abstracts was independently conducted by two authors (C.E.C.S. and M.B.), followed by a full-text reading of studies that met the inclusion criteria. Disagreements were resolved by the third reviewer (M.B.V.). When full-text access was unavailable, the corresponding authors were contacted via email or through the Researchgate platform—www.researchgate.net (accessed on 15 September 2024).

### 2.6. Data Extraction and Analysis

For each included article, the following information was extracted: author, year of publication, country, article type, study design, setting (prison or jail), population (number of inmates, age, gender, and clinical condition or treatment prescribed), clinical services and/or activities provided by pharmacists, processes measures and/or outcomes analyzed, and main results of the studies.

The identified clinical services and/or activities provided by pharmacists were categorized into three main categories [16,17]: ‘Direct patient care’; ‘Medication order review and reconciliation’; and ‘Medication counseling, education, and training’. ‘Direct patient care’ includes serving as a resource on the optimal use of medication in symptom management, optimizing medication regimens, and improving adherence to medication regimens. ‘Medication order review and reconciliation’ involves managing and improving the medication-use process in patient care settings, optimizing medication regimens, and increasing patient safety and pharmacoeconomy. ‘Medication counseling, education, and training’ encompasses providing medication counseling and training to educate staff, patients, caregivers, and families.

Donabedian’s framework for the process dimension was used, which refers to the core activities within healthcare, including diagnosis, treatment, rehabilitation, prevention, and patient education, generally performed by specialized professionals [18]. The outcomes were evaluated considering the Economic, Clinical, Humanistic Outcomes (ECHO) Model [19], which categorizes outcomes into three domains: clinical (e.g., improved disease or symptom control), humanistic (e.g., patient satisfaction and quality of life), and economic (e.g., reduction in healthcare costs) [20].

Two reviewers (C.E.C.S. and B.B.S.) performed the data extraction and analysis using a preformatted spreadsheet in Microsoft Excel, with disagreements resolved by the third reviewer (M.B.V.).

Following the PRISMA-ScR guidelines [14], no methodological quality (risk of bias) assessment was performed as scoping reviews aim to identify all the available evidence and highlight its main characteristics, regardless of the quality of such evidence. The results of this scoping review are presented as a narrative and tabular synthesis.

## 3. Results

### 3.1. Search Results

The electronic search found 894 potentially relevant studies. After removing duplicates and reviewing the titles and abstracts, 58 articles were selected for full-text reading. After careful full-text screening, 27 articles [21,22,23,24,25,26,27,28,29,30,31,32,33,34,35,36,37,38,39,40,41,42,43,44,45,46,47] met the inclusion criteria and were included in the review. Additionally, no relevant studies were identified from searching the reference lists of the included studies or other literature reviews related to the theme. A flowchart of the literature search is shown in Figure 1.

### 3.2. Characteristics of the Articles

Table 1 summarizes the main characteristics of the included studies. All the included articles were published between 1982 and 2023. Most of the articles in this review originated from the United States (*n* = 16; 59%) [21,22,23,24,25,26,27,28,29,30,31,32,33,34,35,36], followed by France (*n* = 7; 26%) [37,38,39,40,41,42,43], Canada (*n* = 3; 11%) [44,45,46], and Ireland (*n* = 1; 4%) [47]. Most of the studies were research/original articles (*n* = 18; 67%) [22,24,26,28,29,30,31,32,33,34,35,37,39,41,42,43,45,46], followed by reports (*n* = 7; 26%) [21,23,25,27,38,44,47] and short/brief reports (*n* = 2; 7%) [36,40]. The authors described the study designs in different ways, but most were descriptive or analytical observational studies; there were no randomized controlled trials. Regarding the settings, 15 studies (55%) addressing prisons (state or federal; long-term duration; for convicted individuals) [21,22,23,24,25,26,27,28,29,30,35,36,41,46,47] and 13 studies (48%) focusing on jails (local—city or county; short-term duration; pre-trial detainees; short sentences) [31,32,33,34,37,38,39,40,41,42,43,44,45]. Among the studies that reported the mean age of patients, the average ranged from 33 to 53 years. Eleven (41%) articles reported exclusively male populations [23,30,32,36,37,38,39,40,42,44,46], while only one (4%) focused on a female-only population [27]. Mixed-gender populations were mentioned in five (19%) articles [31,34,35,41,43], but the proportion of females was low. Furthermore, the most prevalent health issues among the studied populations were mental health conditions and substance use disorders, which were the focus of nine articles (33%) [31,33,36,37,39,42,43,44,47], followed by infectious diseases such as HIV/AIDS and Hepatitis B (HBV) and C (HCV), addressed in five studies (19%) [25,26,29,30,35]. Regarding chronic diseases—excluding mental health conditions—diabetes mellitus was the main focus in four articles (15%) [27,28,32,40], thromboembolism in two (7%) [24,34], arterial hypertension in one (4%) [22], chronic noncancer pain in one (4%) [46], and seizures in one (4%) [23]. Other studies did not report the health conditions of incarcerated individuals or included people with various conditions without focusing on any specific one.

### 3.3. Synthesis of Clinical Services and/or Activities Provided by Pharmacists

The pharmacists carried out several clinical services and/or activities, as detailed in Table 2 and Appendix B. Most articles cited clinical services and/or activities related to ‘Direct patient care’ (*n* = 18; 67%) [21,22,23,24,25,26,28,29,32,33,34,35,36,38,44,45,46,47], involving direct patient contact and active participation in patient-centered care, with a focus on improving outcomes through the optimization of medication regimens. Most of these studies were conducted in the United States and Canada [21,22,23,24,25,26,28,29,32,33,34,35,36,44,45,46]. This was followed by ‘Medication counseling, education, and training’ (*n* = 14; 52%) [22,23,24,25,27,28,30,31,32,34,35,36,40,46], which encompassed patient counseling and education on diseases, treatment regimens, appropriate medication use and adherence, dietary guidance, and the importance of communication with healthcare professionals. Most of these studies were conducted in the United States [22,23,24,25,27,28,30,31,32,34,35,36]. ‘Medication order review and reconciliation’ was cited less frequently (*n* = 9; 33%) [21,22,30,37,38,39,41,42,43]; this involves more indirect patient contact, typically through chart and medical records reviews, order assessment, or healthcare team discussions and meetings, providing technical and clinical support for the team, with a focus on process optimization and medication safety. Most of these studies were conducted in France [37,38,39,41,42,43].

### 3.4. Synthesis of Process Measures and Outcome

The process measures and outcomes identified in the studies are detailed in Table 3. Three articles did not report any outcomes or process measures [21,38,44]. Among the 27 studies, process measures were reported in 18 (67%) [22,23,25,26,29,30,32,33,35,36,37,39,41,42,43,45,46,47], such as pharmacist interventions [26,29,33,35,37,41,45,46], improvements in medication/treatment adherence [23,25,32,36], reductions in the average time physicians spent per patient [22], increases in statin prescriptions [32], reductions in mean daily dose of benzodiazepines [37,39,42], and decreases in the prescription of antibiotics [30] and in the use of nonsteroidal anti-inflammatory drugs [46]. Thirteen studies (48%) included clinical outcomes [22,24,26,27,28,29,32,33,34,35,36,40,47], highlighting the pharmacist’s role in improving health outcomes and treatment effectiveness. Reported clinical outcomes include reductions in HbA_1c_ levels [27,28,32,40], effective control of the International Normalized Ratio (INR) [24,34], reductions in blood pressure and improved hypertension management [22], increased rates of undetectable HIV viral load or favorable viral load response [26,29], high sustained virologic response rate in HCV treatment [35], improvement in mental health symptoms [36], and reach of fully detoxification from methadone [47]. Additionally, five studies (19%) reported humanistic outcomes [22,27,31,40,46], including patient satisfaction [40,46], improved knowledge about medications [31,40], patient empowerment and confidence [27,31], and acceptance of the pharmacist’s services [22]. Ultimately, three studies (11%) reported economic outcomes related to pharmacists’ interventions [22,35,36], particularly physician salary savings [22,36], reduction in medication acquisition costs [22], and lower cost to cure [35].

## 4. Discussion

This scoping review identified a growing number of articles addressing the clinical role of pharmacists in the care of incarcerated individuals in correctional facilities, which exceeded our initial expectations. However, all included studies originated from high-income countries. This may be attributed to persistent challenges in low- and middle-income countries regarding the provision of pharmaceutical care in correctional facilities. These challenges include structural weaknesses such as the absence of pharmacies, non-compliance with legal requirements related to the availability of qualified professionals with technical expertise in medication dispensing, and inadequate adherence to quality and safety standards. Furthermore, there is often a lack of clear guidelines for medication use and storage, as well as ambiguity concerning the health-related responsibilities of governmental authorities [11,48]. In contrast, high-income countries tend to have more robust correctional facilities infrastructures, enabling pharmacists to provide not only logistical support but also clinical services.

Most studies either do not report gender-specific data, focus exclusively on male individuals, or include a disproportionately low number of women. Notably, only one study focused exclusively on incarcerated women, and it addressed health education for diabetes [27]. However, this is not a topic specific to women’s health, although women with diabetes are at higher risk of cardiovascular complications than men with the same condition [49]. This highlights the need for further research on the pharmacist’s clinical role in women’s health, menstrual dignity, menopause, pregnancy, and the management of newborn care in correctional facilities. Other issues that could also be explored in studies involving incarcerated women include the management of substance abuse and sexually transmitted infections. Although the prevalence of drug abuse is higher among women than men, treatment options for female inmates remain more limited [50]. Moreover, incarcerated women are five times more likely to be HIV-positive compared to women in the general population [51].

Most of the studies were conducted in prisons rather than jails. This is expected, mainly because people stay much longer in prisons, which requires continuous monitoring and management of chronic health conditions. In prisons, due to the extended length of stay, pharmacist services can develop more structured programs to monitor and optimize medication use, improve adherence, prevent adverse events, and even provide health education. In jails, higher turnover rates and shorter lengths of stay limit opportunities to provide chronic disease care [32]. However, contrary to this expectation, the study by Lin et al. [32] demonstrated that improvements in glycemic control achieved through pharmacist intervention in a jail setting were comparable to, or even better than, those observed in prison settings. Additionally, within these facilities, pharmacists can play a targeted role in monitoring medications with a narrow therapeutic index [34] and in managing withdrawal associated with substance use disorders [31,33,44].

The clinical role of pharmacists in serving the incarcerated population primarily involves clinical services and/or activities related to direct patient care, especially in the United States and Canada. This aligns with the recommendations of the ASHP, which advocates for pharmacists in correctional facilities to take an active role within the healthcare team responsible for incarcerated individuals, delivering direct patient care aimed at improving health outcomes—particularly within the framework of collaborative practice agreements [6,12]. Another service frequently highlighted in the studies was related to patient counseling and education, further reinforcing the pharmacist’s close involvement in the care of incarcerated patients. This is also emphasized by the ASHP, which recommends that pharmacists provide appropriate counseling to incarcerated patients through the availability of educational materials, direct counseling, or small group meetings, using language adapted to their context, since inmates often have low health literacy [6]. Despite this, one study included in this review, conducted in France in 2010, reported persistent challenges in implementing patient education within the correctional facility, primarily due to limited resources and security-related constraints [38]. This may also explain why the French studies focused more on medication order review.

Despite the promising findings regarding the role of pharmacists in mental healthcare within correctional facilities, more detailed data are needed to better evaluate clinical outcomes and the cost-effectiveness of pharmacists’ interventions in this area, as well as to support the expansion of training opportunities and professional involvement [52,53]. Mental health conditions are risk factors for substance use disorders and the relationship is bidirectional [54]. Given that pharmacists play a crucial role in supporting individuals through substance withdrawal, their work represents a significant contribution to the process of social reintegration [33].

The review demonstrated that pharmacists have a positive impact on clinical outcomes related to HIV and HCV. UNAIDS report highlights that 4.2% of the global incarcerated population is HIV-positive, 15.4% are infected with HCV, and 4.8% with chronic HBV. The incarcerated population has a 7.2 times higher risk of living with HIV than the general population, primarily due to interruptions in treatment during admission, transfer, or release [51], as well as exposure to risk factors like injectable drug use with reused syringes or unprotected sexual activity [50,55].

According to the International Diabetes Federation, in 2021, there were 537 million adults (20–79 years old) living with diabetes worldwide, with projections that this number will rise to 783 million by 2045. In the USA alone, the prevalence of diabetes mellitus in the general adult population was estimated to increase from 9.8% to 12.4% between 1988 and 1994 and 2011–2012, respectively [56]. Rolling et al. found that individuals involved in the criminal justice system were 15% (*p* = 0.015) more likely to receive a diabetes mellitus diagnosis compared to those not involved in the system, with income and level of education being risk factors for both populations [57].

A person with uncontrolled diabetes mellitus is subject to many complications, such as ketoacidosis, retinopathy, neuropathy, nephropathy, and macrovascular complications [58]. Thus, the findings of this review highlight the impact pharmacists have on inmate-patients with diabetes, helping them achieve better disease control, such as lowering HbA_1c_ levels closer to the target for good glycemic control (guidelines recommend a HbA_1c_ < 7.0%) [59,60]. In the four studies included in this review that reported diabetes outcomes, the range of average reduction in HbA_1c_ was from 0.6 to 2.3% points [27,28,32,40]. An average of reduction close or inside this range was found in some literature analyzing pharmacist-led collaborative care of patients with type 2 diabetes in non-correctional settings (literature reduction values of 1.5% [61], and 0.46% [62]).

This scoping review conducted a broad mapping of the services and clinical activities that pharmacists may provide in correctional facilities. As shown in systematic reviews evaluating the economic impact of pharmacist services, particularly in the management of chronic conditions, the inclusion of additional clinical activities within the pharmacist’s role has been associated with cost-beneficial and cost-effective outcomes [63,64]. Based on the findings and topics previously discussed, this review may serve as a foundation for the development of projects and programs aiming to expand the role of pharmacists in such environments. Their involvement contributes not only to improved clinical outcomes for incarcerated individuals but also to economic benefits, such as physician salary savings [22,36], which align with institutional management priorities. Integrating pharmacists may enhance resource efficiency by allowing higher-cost staff to focus on patients with more complex clinical needs [22,23]. In addition, pharmacists also play a key role in providing disease-related education, even in contexts where access to medications is limited [8,9], and in supporting understaffed healthcare teams to offer better health opportunities to this neglected population [65,66].

Although this scoping review presents valuable insights, it is limited by inconsistent methodological approaches and variable presentation formats, as well as the accessibility of results across the included articles. Since scoping reviews do not require a quality assessment of the included studies, this review encompasses broad practices that, while varying in methodological rigor, help illustrate the diverse and multifaceted roles of pharmacists in correctional facilities. For instance, some articles were included despite not reporting any outcomes or process measures [21,38,44]. Other examples can be found in studies that report outcome data but lack essential information necessary for comprehensive analysis. These gaps range from fundamental elements, such as population characteristics, detailed intervention descriptions, and limited process measure and outcome assessment, to more advanced aspects, including the use of rigorous study designs such as randomized clinical trials.

To the best of our knowledge, this is the first scoping review to identify studies reporting on the role of pharmacists and the clinical services and/or activities they provide in correctional facilities. The use of three databases, the inclusion of gray literature, manual searching, and the absence of language restrictions reflect the robustness of the comprehensive search strategy. However, this review has limitations. Twelve potentially eligible studies were excluded due to the inability to retrieve the full texts, despite multiple attempts and strategies to obtain them. Moreover, most of the studies analyzed come from high-income countries, which may limit the applicability of the findings to low- and middle-income contexts.

## 5. Conclusions

This scoping review identified a substantial number of studies highlighting the pharmacist’s clinical role in the care of incarcerated people, particularly in direct patient care and patient education. Mental health and substance abuse, infectious diseases, and diabetes were the most conditions addressed. Moreover, process measures and clinical outcomes were frequently reported. The results of our study highlight the scarcity of research conducted in low- and middle-income countries, particularly studies focused on women’s health—including prenatal and postpartum care—as well as other prevalent conditions in incarcerated populations, such as tuberculosis and syphilis. These represent critical gaps that should be addressed in future research. Targeted studies in low- and middle-income countries are recommended to better understand the barriers and opportunities related to the integration of pharmacists into local correctional health systems. Additionally, research focused on incarcerated women and other relevant health conditions is essential to further demonstrate the role and potential impact of pharmacists in these specific contexts.

## Figures and Tables

**Figure 1 pharmacy-13-00113-f001:**
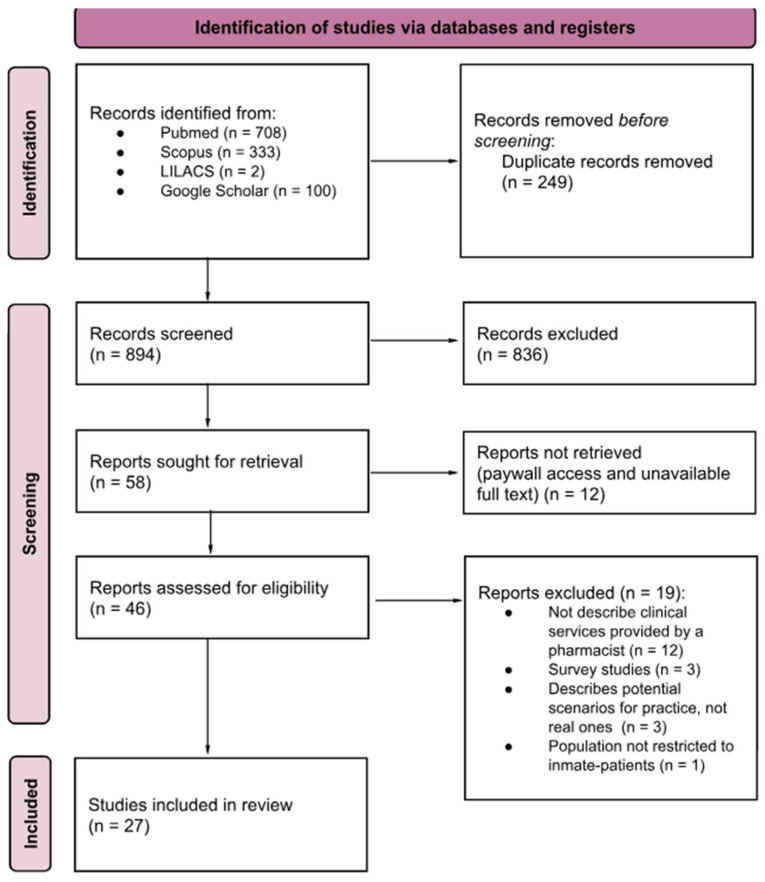
Study selection flowchart through literature search.

**Table 1 pharmacy-13-00113-t001:** Characteristics of the included articles in this scoping review (*n* = 27).

Authors (Year)	Country	Article Type	Study Design	Setting	Number of Inmates	Age	Gender	Condition or Treatment Prescribed
Hadd (1982) [21]	USA	Report	Descriptive qualitative study	Federal Correctional Institution, La Tuna (Texas)—Federal Bureau of Prisons	Inmates accommodated in the main institution (>500) and inmates accommodated in the camp facility (>150)	NR	NR	Acute and chronic diseases (not specified)
Cassidy et al. (1996) [22]	USA	Research Report	Quasi-experimental study	Two state correctional facilities in Texas	151	Chart review clinic: mean age of 44 years. Interview clinic: mean age of 38 years.	NR	Chronic diseases (hypertension, heart disease, diabetes mellitus, oral anticoagulation, hyperlipoproteinemia, epilepsy/seizure disorder, asthma/chronic obstructive pulmonary disease, and infectious disease). Most patients had arterial hypertension
Seals and Keith (1997) [23]	USA	Report	Descriptive study	18 prison units within the Texas Department of Criminal Justice (9 with pharmacist-operated clinics and 9 without)	NR	NR	Men	Anticonvulsant
Mathis and O’Reilly (2010) [24]	USA	Research Article	Descriptive study	Maryland Division of Correction’s Eastern Correctional Institution	12	NR	NR	Warfarin
Badowski and Nyberg (2012) [25]	USA	Report	Descriptive study	All 28 state prisons in Illinois	700	NR	NR	HIV/AIDS
Bingham (2012) [26]	USA	Research Article	Descriptive study	Federal Bureau of Prisons	August 2005, *n* = 58 January 2006, *n* = 135	NR	NR	HIV/AIDS
Barnes et al. (2013) [27]	USA	Report	Descriptive study	Baylor Women’s Correctional Institution	NR	NR	Women	Type 1 and 2 diabetes mellitus
Bingham and Mallette (2016) [28]	USA	Research Article	Descriptive study	Federal Bureau of Prisons	126	NR	NR	Diabetes mellitus
Dong et al. (2017) [29]	USA	Research Article	Retrospective study	USA Federal correctional facilities; Clinician Consultation Center and Federal Bureau of Prisons	34	NR	NR	HIV/HBV/HCV
Long, LaPlant, and McCormick (2017) [30]	USA	Research Article	Descriptive study	Federal Bureau of Prisons (focused on one unidentified institution with a strong active program; eight unidentified institutions were used for comparison)	NR	NR	Men	Antibiotics
Leung, Colyer, and Zehireva (2021) [31]	USA	Original Article	Descriptive study	CHS of Cook County Cook County Jail	60	NR	Men (83%) Women (17%)	Naloxone
Lin et al. (2021) [32]	USA	Research Article	Pre-post study	Twin Towers Correctional Facility and Men’s Central Jail	240	Mean age of 52 years [range = 40 to 64 years]	Men	Type 2 diabetes mellitus, treated solely with oral antidiabetic medications
Muradian et al. (2021) [33]	USA	Research Article	Descriptive study	ADU in the Los Angeles County jail	282	NR	Men	High risk of severe alcohol withdrawal symptoms
Tran et al. (2021) [34]	USA	Research Article	Descriptive study	Los Angeles County jail	116	NR	Men (89%) Women (11%)	Warfarin
Masuda et al. (2023) [35]	USA	Research Article	Nonrandomized retrospective cohort study	Virginia Department of Corrections Facilities	1040	Mean age of 42.7 years [range = 22 to 76 years]	Men (85%) Women (15%)	HCV (genotypes 1 to 6)
Patel (2023) [36]	USA	Brief Report	Descriptive study	Federal Correctional Center Butner	125	<24 years: 1.2%; 24–64 years: 89.3%; >65 years: 9.5%	Men	Schizophrenia, with or without additional mental health conditions, such as bipolar disorder or major depression
Cabelguenne et al. (2007) [37]	France	Original Article	Descriptive study	Jails in Lyon	802	NR	Men	Psychiatric treatment
Harcouët (2010) [38]	France	Report	Descriptive study	*Maison d’arrêt* de Paris–La Santé	1337	NR	Men	Chronic diseases/chronic drug treatments
Lerat et al. (2011) [39]	France	Original Article	Retrospective study	Lyon’s jail	Total, *n* = 473. Control group (before guidelines), *n* = 222. Intervention group (after guidelines), *n* = 251.	Control group: mean age of 33 years. Intervention group: mean age of 35 years	Men	BZD for drug dependence or mental health condition
Davoust et al. (2016) [40]	France	Short Research Report	Prospective observational study	Jail of Marseille	Total, *n* = 30. Medication-focused workshop group, *n* = 15. Control group (patients who chose other workshops, such as those on diet or physical activity), *n* = 15.	Medication-focused workshop group: 49.3 ± 10.8 years. Control group: 48.7 ± 13.9 years.	Men	Type 2 diabetes mellitus
Lalande et al. (2016) [41]	France	Original Article	Retrospective study	*Maison d’arrêt* de Lyon-Corbas and Saint-Quentin-Fallavier Penitentiary Center	NR	The mean age of patients whose prescriptions showed issues was 39 ± 12 years	Both men and women, with a ratio of 8 men for every woman, reflecting an overall female-to-male inmate ratio of 1 to 20 in the prisons mentioned	Various
Cabelguenne et al. (2018) [42]	France	Original Article	Retrospective study	Jails in Lyon	1249	NR	Men	BZD as anxiolytics or hypnotics
Picard et al. (2019) [43]	France	Research Article	Retrospective study	*Maison d’arrêt* de Lyon-Corbas	2011, *n* = 61 2015, *n* = 60	NR	Men (89%) Women (11%)	Antipsychotics
Denning (2011) [44]	Canada	Report	Descriptive qualitative study	Toronto Jail	NR	NR	Men	Methadone
Bhat et al. (2020) [45]	Canada	Original Article	Retrospective electronic chart review	Edmonton Remand Center	518	>18 years	NR	NR
Dawson et al. (2023) [46]	Canada	Original Research	Prospective case series	Correctional Services Canada institutions in British Columbia	53	53 ± 11 years	Men	NSAIDs for chronic non-cancer pain
Cronin, Ryan, and Lyons (2014) [47]	Ireland	Report	Retrospective cohort study	Mountjoy Prison Complex (excluding Dochas Women’s Prison)	416	NR	NR	SSD of MMT

Abbreviations: Acquired Immune Deficiency Syndrome (AIDS), Alcohol Detox Unit (ADU), Benzodiazepines (BZD), Cermak Health Services of Cook County (CHS), Hepatitis B Virus (HBV), Hepatitis C Virus (HCV), Human Immunodeficiency Virus (HIV), Methadone Maintenance Therapy (MMT), Not reported (NR), Nonsteroidal Anti-inflammatory Drugs (NSAIDs), Self-directed Detoxification (SSD), United States of America (USA).

**Table 2 pharmacy-13-00113-t002:** Clinical services and/or activities provided by pharmacists in the articles included in this scoping review (*n* = 27).

Authors	Direct Patient Care	Medication Order Review and Reconciliation	Medication Counseling, Education and Training
Hadd [21]	X	X	
Cassidy et al. [22]	X	X	X
Seals and Keith [23]	X		X
Mathis and O’Reilly [24]	X		X
Badowski and Nyberg [25]	X		X
Bingham [26]	X		
Barnes et al. [27]			X
Bingham and Mallette [28]	X		X
Dong et al. [29]	X		
Long, LaPlant, and McCormick [30]		X	X
Leung, Colyer, and Zehireva [31]			X
Lin et al. [32]	X		X
Muradian et al. [33]	X		
Tran et al. [34]	X		X
Masuda et al. [35]	X		X
Patel [36]	X		X
Cabelguenne et al. [37]		X	
Harcouët [38]	X	X	
Lerat et al. [39]		X	
Davoust et al. [40]			X
Lalande et al. [41]		X	
Cabelguenne et al. [42]		X	
Picard et al. [43]		X	
Denning [44]	X		
Bhat et al. [45]	X		
Dawson et al. [46]	X		X
Cronin, Ryan, and Lyons [47]	X		
TOTAL OF STUDIES (%)	18 (67%)	9 (33%)	14 (52%)

**Table 3 pharmacy-13-00113-t003:** Process and/or outcome measures and main findings on the impact of pharmacists integrated within interdisciplinary teams in correctional facilities (*n* = 24).

Article	Process		Clinical		Humanistic		Economic	
	Variables	Results	Variables	Results	Variables	Results	Variables	Results
Cassidy et al. [22]	Mean time saved per patient by physicians through substituting practitioner time with pharmacist time	In the patient interview clinic: physicians: 10 min; physician assistants: 14 min. In the chart review-only clinic: 4 min.	Systolic and diastolic blood pressure	Reductions were observed in both mean systolic and diastolic blood pressure (*p* = 0.05).	Patient acceptance	Most patients responded positively to pharmacist-led counseling, and none requested to see a physician instead.	Annual direct salary savings	Using a pharmacist for chronic care led to salary savings of USD 67,000 compared to a physician and USD 15,000 compared to a physician assistant.
			Change in hypertension control status	Among 79 patients analyzed: 25 transitioned from uncontrolled to controlled hypertension; 25 maintained controlled blood pressure; 18 remained uncontrolled; 11 shifted from controlled to uncontrolled.			Drug acquisition cost	Decreased by USD 14 per day, corresponding to estimated annual savings of $5110.
Seals and Keith [23]	Medication adherence	Anticonvulsant compliance improved across all prison units, with the greatest gains in units with pharmacist-operated clinics.						
Mathis et al. [24]			Proportion of patients within the target INR range	Increase in patients within target INR range from 50% (6/12) to 66.7% (8/12) after 12 weeks.				
Badowski and Nyberg [25]	Continuity of care during patient transfers between state prisons	Improved						
	Patient safety via multidisciplinary team specialized in infectious disease	Enhanced						
	Prescribing practices and complication management	Improved						
	Pill burden and dosing frequency	Reduction						
	Medication adherence	Improved						
	Identification of drug interactions and toxicities	Improved						
Bingham [26]	Pharmacist interventions	206 interventions performed, including initiation of antiretroviral therapy.	Proportion of patients with undetectable HIV viral load	Increase from 32% to 53%.				
Barnes et al. [27]			HbA_1c_ level	Average HbA_1c_ reduction of 1.6% one year after program initiation.	Patient empowerment and confidence	Inmate empowerment and confidence in diabetes care improved, enabling trainees to educate peers.		
			Proportion of patients with HbA_1c_ reduction	78%				
			Proportion of patients who achieved HbA_1c_ below 8%	22%				
Bingham and Malette [28]			HbA_1c_ level	A mean reduction of 2.3% points was observed, from 10.6% to 8.3%, representing a 22% relative decrease.				
Dong et al. [29]	Pharmacist interventions	Change in the antiretroviral regimen occurred in 87.5% of the cases.	Proportion of cases with favorable viral load response	89% (64%, complete virologic suppression; 25%, significant viral load reduction).				
Long, LaPlant, and McCormick [30]	Antibiotic prescription rate per 1000 inmates	A reduction of 24.6% was observed.						
Leung, Colyer, and Zehireva [31]					Knowledge gain	83.1% stated that the training about naloxone nasal spray kit offered information that they did not have before.		
					Patient empowerment and confidence	After the training, 93.3% felt confident using the naloxone nasal spray kit, and 70% shared the knowledge with family and friends.		
Lin et al. [32]	Medication adherence	Improved	HbA_1c_ level	Reduction from 8.2% to 7.6% (*p* < 0.001). Patients with the highest baseline HbA_1c_ showed greater improvement, with a reduction of 3.1% (*p* < 0.001).				
	Statin prescription	Increased by 50.4%, improving compliance with cardiovascular risk reduction guidelines.						
Muradian et al. [33]	Pharmacist interventions	Pharmacotherapy changes were made for 52% of patients, totaling 180 adjustments.	Mortality	None of the patients died.				
	Transfers and Referrals	48 patients were transferred to an acute care facility, while 163 were referred to a substance use counselor and 73 to a medical and/or mental health provider.						
Tran et al. [34]			Proportion of measures within the target INR range	68% of INR values were within the therapeutic range (target for good control: >65% of INR readings within range).				
			Hospitalizations due to thrombosis or bleeding	None				
Masuda et al. [35]	Pharmacist interventions	Pharmacist follow-up was key for monitoring adverse effects and managing drug interactions. The most common intervention involved proton pump inhibitors (25.7%), which were withheld for 12 weeks or spaced 12 h from the DAA. HMG-CoA reductase inhibitors required adjustments in 5.9% of patients, with treatment suspended or doses reduced to prevent myopathy and rhabdomyolysis.	Cure rate	97%			Cost to cure	USD 23,223/person
			Treatment discontinuation due to adverse events	None				
Patel [36]	Treatment adherence	Of the patients who had previously declined psychiatric medication, 43% consented to begin treatment after participating in the antipsychotic psychoeducation meeting.	Improvement of symptoms	74% of patients experienced stable or improved symptoms of schizophrenia, bipolar disorder, or depression.			Annual salary savings	576 pharmacists’ visits demonstrate an annual salary cost savings of USD 151,000 compared to the cost if the service were provided by a psychiatrist.
	Clinical monitoring performed by the pharmacist	Laboratory tests for narrow therapeutic index medications, AIMS testing, and clozapine REMS (100% of patients); metabolic monitoring (80% completed).						
			Adverse effects	Psychiatric-related movement disorders, as well as other adverse effects, were effectively managed.				
Cabelguenne et al. [37]	Prescription error identified by pharmacist	Most frequent: nonconformities with established standards (30.8%) and drug interactions (22.6%).						
	Pharmacist interventions	Of 2799 prescriptions, 5% needed pharmacist intervention—mainly drug discontinuation (37.6%) and therapeutic monitoring (31.6%)—with 57.1% of interventions accepted.						
	Mean daily dose of BZD	Reduction in DE from 45.9 mg to 33.6 mg per day (*p* = 0.001).						
	Annual number of TU of buprenorphine and Diantalvic^®^	Reduction observed for both buprenorphine (from 18,550 to 8152 TUs/year) and Diantalvic^®^ (from 9600 to 5459 TUs/year).						
Lerat et al. [39]	Percentage reduction in the mean daily dose of BZD in DE	All patients: control group, 46 mg; intervention group, 34 mg (*p* = 0.000), representing a 26% dose reduction. Mental condition: control group, 48 mg; intervention group, 30 mg (*p* = 0.000), reflecting a 37% dose reduction. Drug dependence: control group, 42 mg; intervention group, 40 mg (*p* = 0.800).						
Davoust et al. [40]			HbA_1c_ level	Medication-focused workshop: mean reduction of 1.18%; control group: mean increase of 0.26% (*p* < 0.001).	Patient satisfaction	All participants considered themselves ‘satisfied’ or ‘very satisfied’ with the workshops.		
					Knowledge gain	The average score for identifying their diabetes medication increased by 77.4% (from 2.2 to 4.2; *p* < 0.05), and the score for the rational use of drugs to ensure safety and effectiveness increased by 78.8% (from 2.6 to 4.5; *p* < 0.05) (scale from 0 to 5).		
Lalande [41]	Prescription error identified by pharmacist	Among 18,205 prescriptions, 22.3% had errors, mainly due to missing monitoring, non-compliance, supratherapeutic dosing, and guideline deviations.						
	Pharmacist intervention	78% of the proposed interventions were accepted by prescribers.						
Cabelguenne et al. [42]	Percentage reduction in the mean daily dose of BZD in DE	A 31% reduction, from 42 mg to 29 mg (*p* < 0.001).						
Picard [43]	Clinical monitoring performed by the pharmacist	Between 2011 and 2015, monitoring of clinical parameters in patients on antipsychotics improved significantly. While only blood pressure had over 50% compliance in 2011, four parameters met this threshold in 2015. Most other parameters also showed better monitoring, with more patients rated as having “satisfactory” to “excellent” follow-up. Pharmacist interventions related to therapeutic monitoring rose markedly, from 41 in 2011 to 214 in 2015.						
Bhat et al. [45]	Pharmacist interventions	Pharmacist interventions were provided to 98.6% of patients (median of 3 per patient); 76.1% had therapy changes, and 73.0% were referred to another healthcare professional.						
	Proportion of patients assessed by a pharmacist within 48 h of admission	34.5%						
Dawson et al. [46]	Percentage reduction in oral NSAIDs use	32.4%			Patient satisfaction	58.5% of participants completed the satisfaction survey: 77% agreed that the pharmacist’s visit improved their overall health and well-being; 97% stated that the pharmacist provided education that helped them achieve their therapy goals; and 87% felt that the pharmacist also helped them understand those goals. Regarding assistance with taking medications safely and correctly, 100% acknowledged the pharmacist’s impact, and 93% agreed that the pharmacist helped them understand the purpose of their medications.		
	Percentage reduction in topical NSAIDs use	0%						
	Percentage reduction in the concentration of diclofenac gel (from 10% diclofenac to 2.32% diclofenac)	74%						
	Number of drug related problems identified by pharmacist	153						
	Pharmacist intervention	Pain management interventions were conducted in 96% of patients, with 100% acceptance by physicians.						
Cronin, Ryan, and Lyons [47]	Proportion of patients who reduced methadone dose by ≥ 20 mL	49%	Proportion of patients fully detoxified from methadone	51% (of these patients, 13% either temporarily relapsed or returned to MMT).				

Abbreviations: Abnormal Involuntary Movement Scale (AIMS), Benzodiazepines (BZD), Diazepam Equivalents (DE), Direct-acting antivirals (DAA), Glycated hemoglobin (HbA_1c_), Human Immunodeficiency Virus (HIV), hydroxymethylglutaryl-coenzyme A (HMG-CoA), International Normalized Ratio (INR), Methadone Maintenance Therapy (MMT), Nonsteroidal Anti-inflammatory Drugs (NSAIDs), Risk Evaluation and Mitigation Strategies (REMS), Therapeutic unit (TU).

## Data Availability

No new data were created or analyzed in this study. Data sharing is not applicable to this article.

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
