# Peer review of "Clinical Role of Pharmacists in the Care of Incarcerated People at Correctional Facilities: A Scoping Review"

_pharmacy, 2025, doi:10.3390/pharmacy13050113_

Round 1

Reviewer 1 Report

Comments and Suggestions for Authors

This paper which is described to describe the literature regarding pharmacy services in prisons provide a nice introduction which summarises the topic well. 

Within this, too much space is however provided to criticising the limited review of the subject in 2019.  It is already 6 years old.  Just needs referencing as a ‘basic literature review, up until 2017’ which identified pharmacist roles in correctional settings as including… We can then see if anything has changed in that period and refer back to it in the discussion.  I think this criticism is the author's rationale for their study but they don't need to do that. 

What I struggle to understand with this paper, is the rationale for this paper.  Because someone else did a limited job a few years ago is not a rationale in itself.  Perhaps it is to better understand the literature surrounding the pharmacist role in correctional settings, thereby informing others of potential opportunities to optimise pharmacist input into care?  Also perhaps to describe how services have been evaluated to inform researchers as to what approaches may be appropriate and effective in these environments.

There also may be a need for a more substantial and up to date review due to the increasing automation of medicines supply processes which enables pharmacists to provide more patient-centred services.  

If we don’t know the field, how can we identify gaps as suggested as an aim?

Method is well presented and clear and looks to be appropriate for a scoping review.

The population data in table 1 is dense and may be better summarised in another table as it includes gender, age, condition, medication and as such is hard to read

Similarly, Table 2 is just large chunks of descriptive text which again is hard to make sense of.

I was expecting a table summarising outcomes not just listing the types of outcome used in a column – this information is unhelpful and links back to the purpose of this paper. Is it for researchers or commissioners?  I would rather know the headline result for each study and perhaps secondary result.

The results need three tables –

Table 1 study description with the last column relating to gender and age distribution of incarcerated population for whom the pharmacy service is provided to

Table 2 Service description with columns for the main services at the top and just ticks where provided, so reader can see which services are commonly reported on.  Final column for any service which is outside of the main ones reported on or other services in addition to those ticked

Table 3 Four columns for outcomes (Process, Clinical, Humanistic, Economic) with listed variables for each study under each and main outcome starred, Column for main finding including description of result before and after and p value if available and column for other findings, summarised

The discussion should then be reviewed given the newly presented accessible data but on initial review look to be appropriate.

Minor points

Within this introduction ‘Suffer from’ better written as ‘diagnosed with’

‘Mental disorders’ more commonly and perhaps appropriately ‘Mental health conditions’

The term ‘on’ a medicine denotes both prescribed and taking. We only know the former for sure and therefore I would always prefer the term ‘prescribed’ to be used instead of ‘on’ when describing patients and their medicines.

Author Response

Comments 1: “Within this, too much space is however provided to criticising the limited review of the subject in 2019.  It is already 6 years old.  Just needs referencing as a ‘basic literature review, up until 2017’ which identified pharmacist roles in correctional settings as including… We can then see if anything has changed in that period and refer back to it in the discussion.  I think this criticism is the author's rationale for their study but they don't need to do that. 

What I struggle to understand with this paper, is the rationale for this paper.  Because someone else did a limited job a few years ago is not a rationale in itself.  Perhaps it is to better understand the literature surrounding the pharmacist role in correctional settings, thereby informing others of potential opportunities to optimise pharmacist input into care?  Also perhaps to describe how services have been evaluated to inform researchers as to what approaches may be appropriate and effective in these environments.

There also may be a need for a more substantial and up to date review due to the increasing automation of medicines supply processes which enables pharmacists to provide more patient-centred services.”

Response 1: Thank you for your comment. The last paragraph of the introduction has been rewritten based on your comment and changes are highlighted in blue. If you believe it can still be improved, please let us know.

Comments 2: “If we don’t know the field, how can we identify gaps as suggested as an aim?”

Response 2: Thank you for your comment. One of the main purposes of a scoping review is precisely to explore areas that may not yet be clearly defined. Consequently, identifying knowledge gaps is not only appropriate but also expected within this type of review.  As stated by Peters et al. (2020), scoping reviews are “exploratory projects that systematically map the literature available on a topic, identifying key concepts, theories, sources of evidence, and gaps in the research.” They are particularly useful when addressing complex topics or areas that have not yet been comprehensively reviewed. For these reasons, mapping the literature and identifying research gaps were the objectives of our review.

Reference: Peters MDJ, Marnie C, Tricco AC, Pollock D, Munn Z, Alexander L, McInerney P, Godfrey CM, Khalil H. Updated methodological guidance for the conduct of scoping reviews. JBI Evid Synth. 2020 Oct;18(10):2119-2126. doi: 10.11124/JBIES-20-00167.

Comments 3: “The population data in table 1 is dense and may be better summarised in another table as it includes gender, age, condition, medication and as such is hard to read. Similarly, Table 2 is just large chunks of descriptive text which again is hard to make sense of.

I was expecting a table summarising outcomes not just listing the types of outcome used in a column – this information is unhelpful and links back to the purpose of this paper. Is it for researchers or commissioners?  I would rather know the headline result for each study and perhaps secondary result.

The results need three tables –

Table 1 study description with the last column relating to gender and age distribution of incarcerated population for whom the pharmacy service is provided to

Table 2 Service description with columns for the main services at the top and just ticks where provided, so reader can see which services are commonly reported on.  Final column for any service which is outside of the main ones reported on or other services in addition to those ticked

Table 3 Four columns for outcomes (Process, Clinical, Humanistic, Economic) with listed variables for each study under each and main outcome starred, Column for main finding including description of result before and after and p value if available and column for other findings, summarised.”

Response 3: Thank you very much for your detailed and insightful suggestions. We appreciate your comments regarding the structure and clarity of Tables 1 and 2, as well as your expectations for a more outcome-focused presentation. In response, we have restructured the Results section to include three tables, as proposed, to improve readability and alignment with the article’s objectives:

  • Table 1 now presents the study descriptions, with columns dedicated to the gender and age distribution of the incarcerated population receiving pharmacy services. Additionally, there are specific columns for the number of inmates and their health condition or prescribed treatment.
  • Table 2 summarizes the clinical activities and services provided by the pharmacist as described in the studies, using thematic category columns: ‘Direct patient care’; ‘Medication order review and reconciliation’; and ‘Medication counseling, education and training’, with checkmarks indicating the presence of each activity/service. However, we have included the full table with a qualitative summary of these activities and services as an appendix, as we believe it is important for readers to have access to this information for reference.
  • Table 3 presents the outcomes, variables, and results organized by domain (Process, Clinical, Humanistic, Economic) for each study.

These changes aim to increase the analytical value of the tables and better meet the needs of both researchers and decision-makers. If there are additional adjustments that could further strengthen the manuscript, we would be happy to consider them. Thank you again for your careful feedback.

Comments 4: The discussion should then be reviewed given the newly presented accessible data but on initial review look to be appropriate.

Response 4:  Thank you for your comment. The discussion has been improved and changes are highlighted in blue.

Comments 5: “Within this introduction ‘Suffer from’ better written as ‘diagnosed with’”

Response 5: Thank you for your comment. The change has been made and is highlighted in blue.

Comments 6: “‘Mental disorders’ more commonly and perhaps appropriately ‘Mental health conditions’”

Response 6: Thank you for your comment. The changes have been highlighted in blue.

Comments 7: “The term ‘on’ a medicine denotes both prescribed and taking. We only know the former for sure and therefore I would always prefer the term ‘prescribed’ to be used instead of ‘on’ when describing patients and their medicines.”

Response 7: Thank you for your comment. The change has been made and is highlighted in blue.

Reviewer 2 Report

Comments and Suggestions for Authors

In the manuscript, C. E. C. Silva and coauthors present a comprehensive scoping review on the roles of pharmacists in correctional facilities. The methodology appropriately follows PRISMA-ScR and JBI guidelines, and the findings are supported by clear tables and a helpful narrative synthesis. The review contributes valuable insights into the impact of pharmacists on the health of incarcerated individuals and highlights key gaps in the literature.

However, there are areas that could benefit from clarification, elaboration, or improvement. These are described below:

1. While the authors have done a commendable job compiling relevant studies, much of the information is presented descriptively without deeper synthesis. It often feels as though the data were laid out rather than critically extracted or interpreted.

  • For example, in Table 2 and parts of the Results section, readers encounter long lists of activities or outcomes without clear thematic grouping, comparative insights, or trends.

  • The authors are encouraged to go beyond summarizing and provide more analytical insights into patterns, gaps, or notable differences across studies.

2. The quality of Figure 1 (Study Selection Flowchart) is poor and appears blurry in the PDF. Please provide a higher-resolution image or vector-based figure for clarity.

  • In the flow diagram:

    • “Reports not retrieved (n=12)”: Please clarify the specific reasons these reports were not retrieved (e.g., paywall access, unavailable full text, or language barriers).

    • “Reports excluded (n=19)”: Consider using bullet points to clearly list the reasons for exclusion. Similarly, the first box listing databases searched would benefit from bullet formatting for improved readability.

3. The manuscript inconsistently uses both numerical digits and spelled-out numbers (e.g., 5 articles on line 163 and four articles on line 169). Please adopt a consistent style throughout the manuscript

4. The manuscript uses the terms prison, jail, and correctional services interchangeably. It would be helpful to clarify whether these terms are treated as synonymous or distinct, as they often reflect different durations of incarceration and types of services (e.g., jail = short-term; prison = long-term).

  • Would it be feasible to classify studies based on these facility types? Such stratification might reveal differential trends in pharmacist involvement between short-term and long-term settings and could add depth to the discussion.

5. All included studies are from high-income countries (U.S., France, Canada, Ireland), which the authors acknowledge. However, the implications of this limited scope could be discussed more deeply. For instance, how might pharmacists' roles differ in low- and middle-income countries (LMICs)? What systemic barriers (e.g., legal, financial, training-related) may hinder pharmacist integration in such settings?

6. The manuscript notes a lack of research on incarcerated women, but this point could be emphasized more strongly in both the Results and Discussion. For example, were there any common themes, limitations, or innovative interventions in the single female-focused study that might guide future research?

Author Response

Comments 1: While the authors have done a commendable job compiling relevant studies, much of the information is presented descriptively without deeper synthesis. It often feels as though the data were laid out rather than critically extracted or interpreted.

  • For example, in Table 2 and parts of the Results section, readers encounter long lists of activities or outcomes without clear thematic grouping, comparative insights, or trends.
  • The authors are encouraged to go beyond summarizing and provide more analytical insights into patterns, gaps, or notable differences across studies.

Response 1: Thank you for your careful and constructive feedback. In response to your comment, we have revised the tables and the corresponding sections of the Results to better organize the information into clear thematic categories. Additionally, we have included more interpretative analyses, identifying patterns and discussing any gaps or inconsistencies observed. If you believe there is anything else that could be improved, please let us know. Thank you again.

Comments 2: The quality of Figure 1 (Study Selection Flowchart) is poor and appears blurry in the PDF. Please provide a higher-resolution image or vector-based figure for clarity.

  • In the flow diagram:
    • “Reports not retrieved (n=12)”: Please clarify the specific reasons these reports were not retrieved (e.g., paywall access, unavailable full text, or language barriers).
    • “Reports excluded (n=19)”: Consider using bullet points to clearly list the reasons for exclusion. Similarly, the first box listing databases searched would benefit from bullet formatting for improved readability.

Response 2: Thank you for your comment. We have improved the quality of Figure 1. Additionally, we have incorporated bullet points as suggested and included the reasons why some reports could not be retrieved.

Comments 3: “The manuscript inconsistently uses both numerical digits and spelled-out numbers (e.g., 5 articles on line 163 and four articles on line 169). Please adopt a consistent style throughout the manuscript”

Response 3: Thank you for your comment. We have standardized that numbers up to ten should be written out in full, while numbers greater than ten should be represented using numerals. The changes have been highlighted in blue.

Comments 4: “The manuscript uses the terms prison, jail, and correctional services interchangeably. It would be helpful to clarify whether these terms are treated as synonymous or distinct, as they often reflect different durations of incarceration and types of services (e.g., jail = short-term; prison = long-term).

Would it be feasible to classify studies based on these facility types? Such stratification might reveal differential trends in pharmacist involvement between short-term and long-term settings and could add depth to the discussion.”

Response 4: Thank you for your comment. We have revised the text to use only the more comprehensive term correctional facilities, which is an umbrella term that includes both prisons and jails. In the results section, we included specific information about the type of correctional facility addressed in each study. We also included a discussion about this. The changes have been highlighted in blue.

Comments 5: “All included studies are from high-income countries (U.S., France, Canada, Ireland), which the authors acknowledge. However, the implications of this limited scope could be discussed more deeply. For instance, how might pharmacists' roles differ in low- and middle-income countries (LMICs)? What systemic barriers (e.g., legal, financial, training-related) may hinder pharmacist integration in such settings?”

Response 5: We appreciate the pertinent observation regarding the geographic limitation of the included studies. Indeed, most of the studies analyzed come from high-income countries, which may limit the applicability of the findings to low- and middle-income contexts. We recognize that the role of pharmacists in low- and middle-income countries may differ due to various structural, legal, and financial factors. This is an important gap to be addressed in future research, and we suggest conducting specific studies in these regions to better understand the barriers and opportunities for pharmacist integration in local correctional health systems. We have included this reflection in the Discussion section to enrich the understanding of the generalizability of the results. The changes have been highlighted in blue.

Comments 6: “The manuscript notes a lack of research on incarcerated women, but this point could be emphasized more strongly in both the Results and Discussion. For example, were there any common themes, limitations, or innovative interventions in the single female-focused study that might guide future research?”

Response 6: Thank you for your comment. We expanded the discussion on this topic and highlighted that the only study focusing exclusively on a female population was related to health education in diabetes; that is, it was not specific to women's health, despite studies indicating that the risk of cardiovascular disease is higher in women with diabetes than in men with diabetes. The changes have been highlighted in blue.

Reviewer 3 Report

Comments and Suggestions for Authors
  1. All 27 studies included in the review were from high-income countries (mainly the USA and France), with no studies from low- or middle-income countries. 
  2. 3 studies out of 27 did not report any process or outcome measures.
  3. Include a brief evaluation of study design strength or risk of bias commentary, even qualitatively.
  4. Some studies were summarized with vague or partial descriptions of services/outcomes.
  5. While the discussion is informative, the manuscript lacks a clear "Recommendations" or "Implications for Practice" section.

Author Response

Comments 1: Include a brief evaluation of study design strength or risk of bias commentary, even qualitatively.

Response 1: Thank you for your comment. Following the PRISMA-ScR guidelines, no methodological quality assessment was performed as scoping reviews aim to identify all the available evidence and highlight its main characteristics, regardless of the quality of such evidence. However, we have included a paragraph in the discussion section addressing methodological issues and general impressions regarding the quality of the studies. The changes have been highlighted in blue.

Comments 2: Some studies were summarized with vague or partial descriptions of services/outcomes.

Response 2:  Thank you for your comment. The results section was revised for a clearer presentation and description of the services and outcomes. The changes have been highlighted in blue.

Comments 3: While the discussion is informative, the manuscript lacks a clear "Recommendations" or "Implications for Practice" section.

Response 3: Thank you for your comment. We have included a paragraph in the discussion section addressing implications for practice. The changes have been highlighted in blue.

Round 2

Reviewer 1 Report

Comments and Suggestions for Authors

Thanks for the revisions. End paragraph and tables 1 & 2 are much clearer.  Could you please however revise Table 3, so the results column (and others where possible) more succinctly summarises the main result from each paper without lots of text.  Lots of words are being used such that the results cannot be easily seen without a lot of reading and as a result the table is far too long.  Where there are secondary outcome perhaps these can be as a legend underneath the table with a symbol to link them.  

Author Response

Comments 1: “Thanks for the revisions. End paragraph and tables 1 & 2 are much clearer.  Could you please however revise Table 3, so the results column (and others where possible) more succinctly summarises the main result from each paper without lots of text.  Lots of words are being used such that the results cannot be easily seen without a lot of reading and as a result the table is far too long.  Where there are secondary outcomes perhaps these can be as a legend underneath the table with a symbol to link them.”

Response 1: We thank the reviewer for the constructive comments and for acknowledging the improvements made to the manuscript, including the end paragraph and Tables 1 and 2. In response to the suggestion regarding Table 3, we have revised the results column to provide a more concise summary of the main findings from each study, reducing the amount of text and improving readability. We believe these changes make the table clearer and more accessible, and we appreciate the reviewer’s helpful guidance in this regard. As the authors of the included articles did not distinguish between primary and secondary outcomes, all reported outcomes are presented in Table 3. Additionally, since the purpose of a scoping review is to broadly explore the existing literature, we did not predefine the outcomes of interest. Our strategy was to include all outcomes reported in the selected studies, in order to map what has been investigated so far on the topic. If you have any further suggestions, please let us know.

Reviewer 2 Report

Comments and Suggestions for Authors

The reviewer appreciates the efforts the authors made to improve the manuscript. The quality and content have been significantly improved. 

My only concern is the formatting of Table 3. Please make sure it is formatted correctly in the final version. 

Author Response

Comments 1: “The reviewer appreciates the efforts the authors made to improve the manuscript. The quality and content have been significantly improved. My only concern is the formatting of Table 3. Please make sure it is formatted correctly in the final version.” 

Response 1: We sincerely thank the reviewer for the positive feedback and valuable suggestions. We are glad to know that the quality and content of the manuscript have been significantly improved. Regarding Table 3, we carefully reviewed and corrected its formatting to ensure clarity and consistency in the final version. If you have any further suggestions, please let us know.

Reviewer 3 Report

Comments and Suggestions for Authors

The manuscript is improved and the comments addressed  

Author Response

Comments 1:  “The manuscript is improved and the comments addressed.”

Response 1: Thank you for your feedback. We are glad to hear that the manuscript has improved and that the comments have been addressed satisfactorily.